# Psychosocial Work Conditions as Determinants of Well-Being in Jamaican Police Officers: The Mediating Role of Perceived Job Stress and Job Satisfaction

**DOI:** 10.3390/bs14010001

**Published:** 2023-12-19

**Authors:** Kenisha Nelson, Andrew P. Smith

**Affiliations:** 1Social Sciences Department, University of Technology, Jamaica, Kingston 12402, Jamaica; knelson@utech.edu.jm; 2Psychology Department, Cardiff University, Cardiff CF10 3AT, UK

**Keywords:** police, Jamaica, stress, work support, job demands, job satisfaction, well-being, physical health, mediation

## Abstract

Policing is considered a high-stress occupation due to the demanding nature of the job. Addressing police stress requires a detailed understanding of how psychosocial risk factors influence various aspects of their well-being. Moreover, an analysis of the direct effects of work conditions and the mediating effects of cognitive appraisals is also warranted. Using the Demands–Resources–Individual Effects (DRIVE) model of work-related stress this study investigated the direct effects of work conditions on well-being and also examined the intermediate role of perceived job stress and job satisfaction in the relationship between work conditions and well-being. Five hundred and seventy-eight police officers from the Jamaica Constabulary Force (J.C.F.) completed the questionnaire. Data were analyzed using hierarchical regressions and the Hayes Process tool for mediation analysis. Both perceived job stress and satisfaction mediated the relationship between work conditions and general physical health. Perceived job stress was an indirect pathway through which work conditions influenced psychological distress, whereas job satisfaction was not a significant mediator. In contrast, job satisfaction mediated the relationship between work conditions and positive well-being, but perceived job stress did not. These findings provide significant evidence for periodically monitoring and auditing perceptions of stress and job satisfaction, as they are likely precursors to subsequent health problems.

## 1. Introduction

Policing is considered a high-stress occupation due to the demanding nature of the job. Along with stressors frequently found in other jobs, police work also involves an increased risk of danger and unpredictable situations. The inherent hazards associated with policing imply that police officers experience a high stress level, exceeding that of most other occupations. Not surprisingly, a growing number of studies have linked police work to poorer physical and psychosocial well-being [1,2,3]. Police officers also show higher mortality rates than the general population [4].

Police officers are tasked with the dangerous job of maintaining law and order, and the organization’s success depends on its officers’ effective functioning. For instance, in small Caribbean states such as Jamaica, policing is increasingly challenging due to unrelenting high crime rates. Economic challenges mean police organizations are typically underfunded and under-resourced. Furthermore, in the Jamaican context, police operate in an environment with strained and antagonistic police–citizen relations combined with heavy scrutiny from international and local human rights advocates. Resource constraints and pressure to perform under tight scrutiny in a high-crime environment no doubt exacerbate the stress and related problems experienced by these police officers. Consequently, developing a better understanding of the psychosocial risk factors within this work environment is essential to help monitor the well-being of police officers and manage the impact on performance.

While several studies have shown that work conditions directly influence personal experiences of physical and psychological strain [2,3,4,5,6], there is little research on whether subjective appraisals, such as perceived job stress and job satisfaction, may affect this relationship. These affective variables reflect how the individual subjectively feels about a potential environmental stressor they have encountered and act as precursors to health-related problems [7]. Studies have identified associations between work conditions, perceived stress, and job satisfaction [8]. Other research has found associations between job stress and satisfaction and health-related outcomes [9]. These results suggest that job stress and satisfaction mediate the relationship between work conditions and well-being outcomes. Furthermore, within the police literature, mediated roles of job appraisals have been supported, at least in part, in a previous study using a relatively small sample of Jamaican police officers [10] and a study of British police [11]. The current study aimed to investigate the direct effects of work conditions on the well-being of Jamaican police officers and examined the mediating role of perceived stress and job satisfaction in these relationships.

To examine these relationships, the present study used the Demands–Resources–Individual Effects (DRIVE) model [12,13] as the theoretical framework. The DRIVE model adopts elements of contemporary occupational stress perspectives while accounting for the influence of subjective appraisals of the stress experience. Specifically, the first component of the model relates to work conditions (e.g., job demands, control, and support) and well-being outcomes (anxiety, depression, physical health, and positive affect) and is based on the direct effects of positive and negative work conditions on outcome measures. Secondly, to represent the subjective appraisal process analogous to the appraisal stages of transactional models, the original model included cognitive appraisals, such as perceived job stress as a mediator in the relationship between work conditions and well-being outcomes. Later versions of the DRIVE model have included other subjective appraisals, such as job satisfaction. A further feature of the model is the inclusion of important individual characteristics such as coping styles. The model, therefore, allows analysis of the direct effects of work and individual factors and the mediating effect of the cognitive appraisals. Evidence supporting the relationships proposed by the model is given in the following sections.

### 1.1. Well-Being

Well-being is a complex construct. The literature has no general agreement on specific well-being predictors or outcomes. From one perspective, “well-being” is generally conceptualized as a state of optimal functioning and experience [14]. That is, well-being refers to a positive psychological state derived from two standpoints: the hedonic approach, which suggests well-being involves pleasure or happiness, and the eudaimonic approach, which suggests well-being consists of fully functioning and realizing one’s potential [14]. However, other approaches encompass a broader conceptualization of this construct [13,14]. For instance, operational definitions in the literature cover various outcomes, including psychological, emotional, or mental concerns and physical states [15,16].

Well-being is likely to reflect a combination of multiple factors, potentially interacting to affect individuals in a complex manner. For instance, Hart and colleagues [17,18], in their well-being framework, suggest that an individual’s overall perceived quality of life includes affective, cognitive, and somatic health components. The affective component is defined by positive (e.g., positive affect, psychological morale and well-being) and negative (e.g., negative affect, psychological distress and ill-being) dimensions, whereas the cognitive component is associated with satisfaction with various life domains [19,20]. Acknowledging that well-being is a multifaceted construct has led to the use of several measures that reflect its components. For this research, negative and positive components of well-being and indicators of general physical health were considered.

### 1.2. Psychosocial Work Conditions and Well-Being among Police Officers

Numerous studies have sought to identify the most significant sources of stress in policing. There are two generally accepted specific sources of police stress: those related to the nature of the organization (e.g., insufficient personnel, inadequate resources, bureaucratic systems) and stressors inherent in the operational duties of policing (e.g., threat of being injured/killed, insults/aggression from the public, high-speed car chases). Both aspects of the job have been linked to adverse psychosocial outcomes. However, most studies show organizational stressors to be more important than operational experiences in predicting physical and psychological health outcomes.

In several cross-sectional studies of Australian police officers, Hart and colleagues [17,18], using measures of police hassles and uplifts, found that organizational factors were the strongest determinants of psychological distress and perceived quality of life.

A longitudinal study of police recruits from New Zealand [21] evaluated the impact of traumatic and organizational experiences on psychological distress. While both job components were associated with distress one year later, organizational stressors had the most substantial effect. Another longitudinal study of urban US police officers [22] found that routine police work in the first year of police service in officers with no previous mental health symptoms was significantly associated with depression at 12 months of follow-up. Similarly, van der Velden, Kleber, Grievink, and Yzermans [23] conducted a longitudinal study of Dutch police officers, examining the relationship between the frequency of exposure to aggression, organizational stressors and life events, and mental health outcomes. The findings indicated that, while the frequency of confrontations with physical aggression was not associated with mental health problems at follow-up, organizational stressors were predictive of mental health issues at baseline and follow-up.

Research in police populations has also shown associations between organizational variables from contemporary stress models and well-being outcomes. Using a small sample of Canadian police officers, Janzen, Muhanjarine, Zhu, and Kelly [24] were among the first groups to test this occupational group’s effort–reward imbalance model of well-being. The results showed that high effort, low reward, and over-commitment were significantly associated with psychological distress. This finding has been confirmed in subsequent studies [6,25]. Other researchers [4,5] have found that high demand, low control, and low social support were significant predictors of psychological distress. In a study of Italian police officers, Gabarino et al. [7] tested the additive effect of components of the DCS and ERI models after controlling for demographic and personality variables. Control, support, and all three dimensions of ERI (i.e., effort, reward, and over-commitment) were associated with depression. Support and reward were associated with lower anxiety levels, and high effort increased anxiety levels.

Houdmont, Randall, Kerr, and Addley [26] used the HSE Management Indicator Tool to assess police officers’ perceptions of work conditions and their association with psychological distress. The measure, developed by the UK Health and Safety Executive (HSE), consists of components taken from the DCS model and other dimensions of the psychosocial work environment (i.e., demand, control, managerial support, peer support, relationships, role clarity, and change). Houdmont and colleagues studied a large group of UK police officers and found that four psychosocial dimensions of the HSE Management Standard tool—demands, control, relationships, and role clarity—were significant predictors of psychological distress. Based on the above literature, the following hypothesis was tested:

**Hypothesis** **1.***Psychosocial work conditions will predict well-being outcomes in police officers*.

### 1.3. Perceived Job Stress as a Mediator

The definition of stress has evolved from including relatively simple components to more intricate relationships among these elements [27]. Early models narrowly focus on stressful stimuli and responses and ignore individual processes [28,29]. As Lazarus [28] suggests, work conditions alone cannot explain the stress process. Consequently, more contemporary authors conceptualize stress as a more dynamic process, arguing that it is essential to consider the interaction between individuals and their environment. This approach recognizes the role of psychological processes, such as perception, cognition, and emotion [29]. To represent this subjective appraisal component, Mark and Smith [13], in the DRIVE model, proposed that affective perceptions, such as perceived job stress, mediate the relationship between work characteristics and health outcomes. In other words, a psychosocial stressor will not transmit its effect on outcomes if it is not perceived as stressful.

While research has examined conditions of work that result in perceived job stress on one hand and the relationship between perceived job stress and health outcomes on the other hand, there has been little attempt to examine the mediating effects of perceived job stress, particularly in police studies. Exceptions include Oliver et al.’s [11] findings on the mediating role of perceived job stress in the relationship between work demands and well-being. Similarly, one police study showed that perceived job stress mediates the relationship between work conditions and work-related outcomes, such as job retention [30]. Research in other occupational groups also supports the mediating role of perceived job stress between work conditions and well-being outcomes [19,31]. Based on the above literature, the following hypothesis was tested:

**Hypothesis** **2.***Perceived job stress will mediate the relationship between psychosocial work conditions and well-being outcomes*.

### 1.4. Job Satisfaction as a Mediator

Job satisfaction has been commonly defined as “the pleasurable or positive emotional state resulting from the appraisal of one’s job or job experiences” [32]. Job satisfaction and other occupational and personal outcomes are typically considered outcome variables. It has generally been shown that work conditions strongly influence police job satisfaction. Specifically, core work conditions from the Job Diagnostic Survey (JDS), namely, skill variety, task identity, feedback, and autonomy, are significantly correlated with levels of job satisfaction [33,34,35]. Additionally, positive work factors such as perceived control over one’s job, role clarity, promotion opportunities, job security, and innovative opportunities increase job satisfaction levels in police officers. In contrast, negative work conditions such as working on shifts, inadequate pay, organizational changes, and job demands/challenges reduce job satisfaction [36,37,38,39].

In summary, when examined as an outcome variable, there are established associations between job satisfaction and work conditions [34,38,39]. When job satisfaction is an antecedent variable, it has been significantly associated with health-related outcomes [39]. These two types of associations suggest that job satisfaction probably mediates between work conditions and well-being outcomes. However, few police studies have attempted to test this mediating effect directly. Based on the above literature, the final hypothesis tested was:

**Hypothesis** **3.***Job satisfaction will mediate the relationship between psychosocial work conditions and well-being outcomes*.

### 1.5. Key Variables in the Present Study

The independent variables used in the present study were based on the DRIVE model. Details of the measuring instruments are given in Section 2. These variables were:Job demands;Control;Supervisor Support;Colleague Support;Effort;Reward;Overcommitment;Bullying;Role conflict;Lack of consultation on change.

In addition, the DRIVE model allows for job-specific variables to be included, and the novel variable used here was victimization.

Based on the well-being process model, the outcome variables were those shown below. Again, details of the measuring instruments are given in Section 2.

Positive well-being (happiness, life satisfaction, positive affect);Negative well-being (negative affect, anxiety, and depression);Physical health.

The mediators were perceived job stress and job satisfaction. The analyses used demographic and occupational information (e.g., rank, years of service) as control variables.

## 2. Materials and Methods

### 2.1. Participants

The participants consisted of active-duty policemen and policewomen who were working in various police divisions and units within the Jamaica Constabulary Force (JCF). The target groups were officers between the constable and inspector ranks. These are police officers referred to as “rank and file” and who largely carry out the day-to-day policing duties. Police officers above inspector ranks were not included as they tend to perform more administrative or managerial duties, are much smaller in numbers, and would have been less easy to access. To recruit participants, a letter was sent to the Jamaican Police Commissioner asking for the study to be advertised. The extent of this, and hence the number of potential participants, was not known, which means that a response rate cannot be calculated. The total sample of the study consisted of 578 police officers. Demographic and occupational statistics for the sample are shown in Table 1.

### 2.2. Measures

Several constructs in this research were assessed using single-item measures. Single-item use is not new in academic research, and the approach has been applied to various research constructs. If researchers are to assess multiple factors in occupational stress and well-being research, they must include multiple measures. This would involve several multi-item scales and a lengthy questionnaire, which can have practical implications related to response time and burden, particularly in work settings. Further, the additional demands placed on the participant are likely to result in lower response rates and increased attrition [40], all of which can impact the validity of responses and affect the generalization of research findings. Therefore, researchers must consider alternative approaches to reduce the frustration and fatigue that come with longer instruments while not limiting the number of measured constructs. Researchers have found support for using single-item measures, and some items have gained respectability in the research literature [40].

#### 2.2.1. Psychosocial Work Conditions

The occupational variables developed and validated by Williams [19] and Williams and Smith [20] were used in this study. Work-related variables consisted of items derived from dimensions of the Demand–Control–Support [41] and Effort–Reward Imbalance [42] models as well as additional items taken from the HSE Management Indicator Tool that are not accounted for in the former two models. The latter is the Management Standards for Work-Related Stress introduced by the UK Agency of Health and Safety Executive (HSE) to assess organizational psychosocial risks. The original instrument consists of 35 items that tap into six constructs, some reflective of the dimensions of the DCS model. The DRIVE model included generic work conditions but also allowed for the inclusion of specific sample-relevant measures. In this study, victimization was added to the model.

#### 2.2.2. Victimization

Participants were asked to indicate how often they had been subjected to aggression and violence while on duty over 12 months. Four types of victimization were examined using five items: verbal threat, physical assault (two items), threat with a weapon, and assault with a weapon. Items were measured on a five-point scale ranging from 0 (never) to 4 (more than 15 times). The sum of the five items constitutes the indicator of victimization. This measure has been used in previous studies and has satisfactory psychometric properties [43,44].

#### 2.2.3. Perceived Job Stress

Single-item measures of perceived stress have been used widely in the literature on workplace stress and shown to be an adequate measure of the construct [45]. A similar item was used in this study: Overall, how stressful is your job? was evaluated along a 10-point Likert-type scale, from 1 (not at all stressful) to 10 (very stressful).

#### 2.2.4. Job Satisfaction

The Warr–Cook–Wall [46] job satisfaction scale requires participants to indicate their satisfaction with 15 job items and a global measure of job satisfaction. Each item was rated on a 7-point Likert scale, from 1 (very dissatisfied) to 7 (very satisfied). The measure assesses both intrinsic and extrinsic job satisfaction. The sum of the items on this scale provided an overall measure of job satisfaction, such that higher scores represent high job satisfaction.

#### 2.2.5. Psychological Well-Being

Items from the Well-Being Process Questionnaire [20] were used to assess well-being (i.e., anxiety, depression, happiness). For example: On a scale of one to ten, how depressed would you say you are in general (e.g., feeling ‘down’, no longer looking forward to things or enjoying things that you used to)? Items had a response scale from 1 (not at all depressed) to 10 (extremely depressed).

A modified version of the Warwick–Edinburgh Mental Well-being Scale [47] used a 5-item scale to assess positive psychological well-being. An example of an item was: I have been feeling in good spirits (e.g., I feel good about myself and confident in my abilities). The items had response scales from 1 (disagree strongly) to 10 (agree strongly).

#### 2.2.6. General Physical Health

A single item, similar to measures used in previous research [45], was used to assess the overall health status of respondents. The item was the following: Over the past 12 months, how has your general physical health been? Respondents indicated their rating on a response scale from 1 (extremely poor) to 10 (extremely good).

#### 2.2.7. Demographic and Occupational Variables

The socio-demographic profile of the participants was determined by asking them to respond to a series of questions defining their demographic characteristics and employment details. These questions included identifying participants’ gender, age, relationship status, rank, level of education, and years of service. These are the standard variables used in most studies of police stress.

### 2.3. Analysis Strategy

The data were analyzed using the Statistical Package for Social Sciences (SPSS) computer program, Version 20.0. Principal Component Analysis (PCA) was first used to reduce the number of items to meet the multivariate analysis requirements, as Tabachnick and Fidell [48] recommended. PCA was performed to derive clusters of associated single-item measures of psychosocial work conditions and psychological well-being outcomes. A similar approach of combining variables has been taken in previous research [13]. Smith and colleagues [13], in assessing multiple constructs in a well-being model, used a “combined effects” approach that combined multiple dimensions of work, including components of DCS, ERI, and HSE models, into measures of work conditions (e.g., demand, role, control) and job appraisals (e.g., reward, satisfaction with supervisor, satisfaction with peers). Broad outcome measures were also created, namely negative mental well-being (e.g., negative mood, depression, anxiety); physical health (e.g., physical health symptoms, fatigue); and positive mental health (e.g., positive mood, happiness). The authors surmise that this approach reduces the number of variables into manageable clusters and reduces the possibility of chance effects in subsequent analysis. Their study also suggests that combined factor scores were the best predictor of outcomes. The independent and dependent variables are shown in Section 3 after the factor analyses that define these variables.

### 2.4. Mediation Analyses

To test hypotheses 2 and 3 (whether perceived job stress and job satisfaction mediate the relationship between psychosocial work conditions and well-being), mediation analysis was performed using the Hayes PROCESS tool [49]. Mediation analysis was performed using a base model including work factors and adjusting for demographic variables. The bias-corrected bootstrap confidence intervals for the indirect effects based on 1000 bootstrap samples are considered significant when the confidence intervals do not contain zero.

## 3. Results

### 3.1. Demographic and Occupational Characteristics of the Sample

Seventy-four per cent of the sample were male. A total of 62.5% were constables, 22% corporals, 11% sergeants, and 5% inspectors. The average time in the police force was ten years (SD = 8.20). A total of 43% served for five or fewer years, 27% for 6–12 years, and 31% served for 13 or more years. Participants’ ages ranged from 20 to 63 years old, with a mean age of 33 years (SD = 8.53). A total of 35% were 28 years or younger, 34% were between ages 29 and 35, and 31% were 36 or over. Just over half of the participants had at least a secondary level education (57%), and the rest had some form of post-secondary education. Most police officers reported being in a relationship (45%), 28% were married, 21% were single, and 6% were separated, divorced, or widowed.

### 3.2. Factor Analysis

PCA performed on the psychosocial work items revealed three components: negative work conditions (Eigenvalue: 1.79; % of the variance: 17.90), positive work conditions (Eigenvalue: 1.06; % of the variance: 10.60) and work support (Eigenvalue: 2.41; % of the variance: 24.09). Items used as indicators of psychological well-being were also subjected to PCA. Two components were revealed and labelled psychological distress (Eigenvalue: 1.14; % of the variance: 14.21) and positive well-being (Eigenvalue: 3.77; % of the variance: 47.1). These results confirm previous findings and reliability analyses show Cronbach alphas >0.8.

### 3.3. Direct Relationships

Hierarchical regression analyses were used to examine the associations between work conditions and mediator variables (perceived job stress and job satisfaction) and work characteristics and well-being outcomes while adjusting for gender, rank, marital status, and job tenure. Work conditions, the independent variables, were negative work conditions, positive work conditions, work support, and victimization. The outcome variables were psychological distress, positive well-being, and physical health.

In the first set of regression analyses, several significant relationships between work conditions and perceived job stress and job satisfaction were found. The results are shown in Table 2. The analysis showed that all work conditions significantly predicted perceived job stress except for work support, with negative work conditions being the most crucial variable by beta weight. Job satisfaction showed significant negative relationships with negative work conditions and victimization and positive relationships with work support and positive work conditions. Regarding the demographic variables, job tenure was positively associated with perceived job stress. The rank of police officers was positively associated with job satisfaction and there was a weaker but significant negative association with relationship status and job satisfaction. The models accounted for 22% and 33% of perceived job stress and job satisfaction, respectively.

Results from the second set of regression analyses showed that negative work conditions, victimization, and work support were significantly associated with psychological distress. Positive well-being was significantly associated with negative work conditions, positive work conditions, and work support. Negative work conditions, positive work conditions, and victimization were associated with general health. Job tenure was also positively associated with general health, but no significant relationships were found with gender, relationship status, and rank. No demographic variables were significantly associated with psychological distress and positive well-being. The regression models accounted for 23%, 17%, and 15% of explained variance in distress, well-being, and general health, respectively.

### 3.4. Mediation

These analyses tested the relationships shown in Figure 1.

#### 3.4.1. Psychological Distress

While there were significant indirect effects via perceived job stress, indirect pathways through job satisfaction were not significant for the relationship between work conditions and psychological distress. There was a significant indirect effect of negative work conditions, *b* = 0.02, 95% CI [0.00, 0.05], victimization, *b* = 0.13, 95% CI [0.01, 0.34], and positive work conditions, *b* = −0.01, 95% CI [−0.03, −0.00] on psychological distress through perceived job stress but not for work support, *b* = −0.00, 95% CI [−0.01, 0.01]. Mediation effects are summarized in Table 3.

#### 3.4.2. Positive Well-Being

The indirect effect of perceived job stress and job satisfaction on positive well-being was the reverse of those seen for psychological distress. That is, while there were significant indirect effects via job satisfaction, indirect pathways via perceived job stress were not significant in the relationships between work conditions and positive well-being. As shown in Table 4, negative work conditions, *b* = −0.05, 95% CI [−0.09, −0.01], work support, *b* = 0.04, 95% CI [0.01, 0.08], positive work conditions, *b* = 0.06, 95% CI [0.02, 0.13], and victimization, *b* = −0.33, 95% CI [−0.77, −0.08] all indirectly influenced positive well-being through job satisfaction.

#### 3.4.3. General Physical Health

As shown in Table 5, statistically significant indirect effects through perceived job stress for three of the four work conditions on general physical health were observed, while all four work conditions affected general health via job satisfaction. Results show that negative work conditions, *b* = −0.01, 95% CI [−0.02, −0.00], positive work conditions, *b* = 0.00, 95% CI [0.00, 0.01], and victimization, *b* = −0.05, 95% CI [−0.13, −0.01] indirectly influenced general health through perceived job stress. Similarly, job satisfaction acted as an indirect pathway through which negative work conditions, *b* = 0.01, 95% CI [−0.02, −0.00], work support, *b* = 0.01, 95% CI [0.00, 0.02], positive work conditions, *b* = 0.01, 95% CI [0.01, 0.03], and victimization, *b* = −0.08, 95% CI [−0.17, −0.02] influenced general health.

Using the PROCESS tool, it is possible to test whether one indirect effect statistically differs from another in a parallel model. Given the fact that negative work conditions, positive work conditions, and victimization all influenced general health through both perceived job stress and job satisfaction, further analyses were conducted to estimate whether the indirect effects were significantly different from each other. The macro for SPSS created by Hayes [49] generates an output that estimates the pairwise comparison by subtracting the specific indirect effect through one mediator (e.g., perceived stress) from the indirect pathway through the second mediator (e.g., job satisfaction). Similar to simple mediation models, bias-corrected bootstrap intervals are estimated for pairwise comparisons between the specific indirect effects. A confidence interval that does not contain zero indicates that the two specific indirect effects are statistically different. Examination of the indirect effect pairwise contrast showed that the indirect effects of negative work conditions, *b* = 0.00, 95% CI [−0.01, 0.01], positive work conditions, *b* = −0.01, 95% CI [−0.03, 0.00], and victimization, *b* = 0.03, 95% CI [−0.07, 0.13] on general health through perceived job stress were no different from the indirect effect through job satisfaction.

## 4. Discussion

There have been continuous calls by police stress researchers to improve our understanding of stress–strain relationships by examining specific pathways through which variables affect each other [50,51]. This argument guided the current research. A model framed within transactional theories of stress was used to investigate work stress and well-being in Jamaican police officers. This paper examined the direct effects of work conditions and mediation effects with perceived job stress and job satisfaction as indirect pathways through which work conditions affect the outcomes.

### 4.1. Direct Relationships

All psychosocial work conditions strongly contributed to psychological distress, though overall relationships between work conditions, positive well-being, and general health were relatively weaker. Regarding significant independent predictors, the results were mixed, which confirms previous research. Negative work conditions were consistent in predicting all dimensions of well-being, though the relationship was stronger for psychological distress. Consistent with previous studies [7,26,30], these results suggest that the negative aspects of the work environment, such as high demands mixed with difficulty withdrawing from work obligations and feeling that one is not consulted on decisions, can be associated with poorer well-being and health outcomes. In contrast, the relationship between victimization and well-being outcomes was weaker. Being exposed to multiple incidents of assault or violence was associated with general health and psychological distress but not positive well-being. The weaker influence of this “operational” variable supports the body of literature that finds organizational factors have a more substantial influence on well-being measures [21,23] compared to operational stressors.

Work support was most important for positive well-being and, to a lesser extent, psychological distress. However, support from work sources was not associated with general health. The weak influence of work support on personal health in the current study is inconsistent with previous police research [7,26]. Current findings suggest that other positive features of the work environment, such as reward opportunities and job control, are associated with positive well-being and general health but not psychological distress. Previous findings also suggested that decision-making authority and other reinforcing elements of the job are essential for the Jamaican police [50]. Evidence for the predictive capacity of these variables is also demonstrated in prior police research. For instance, Gabarino et al. [7], in their study of Italian police officers, found strong support for the influence of similar dimensions from the DCS and ERI models. It may also be possible that the influence of positive work variables may vary depending on what dimensions of distress are being measured. For example, in previous research by Nelson and Smith [10], work support and positive work conditions were significantly associated with depression but not with anxiety. In sum, the current research suggests that conditions of work exert a relatively strong influence on police officers’ well-being. However, some conditions appear more variable than others, and relationships may differ depending on the type of well-being being investigated. Addressing the modifiable conditions within the work environment can help reduce adverse outcomes while improving health, but understanding the nuances of the relationships between specific work factors and outcomes is also essential for targeted interventions.

### 4.2. Indirect Effects

While work conditions were directly related to personal well-being, they at least, in part, exert their influence through the cognitive pathways of job appraisals. Interestingly, perceived job stress was an indirect pathway through which work conditions (except work support) influenced psychological distress, but job satisfaction was not. In contrast, job satisfaction mediated the relationship between work conditions and positive well-being, but perceived job stress did not. However, perceived job stress and satisfaction played a similar intermediate role in the relationship between work conditions and general physical health. Support for a mediated pathway through these variables has also been demonstrated in previous police research [30,51,52]. Moreover, this finding partially supports the cognitive-relational hypothesis of occupational stress models such as that proposed in the DRIVE model [12,13].

### 4.3. Strengths and Limitations of the Research

This research has provided valuable insights into the stress process by presenting evidence to support the direct effects of work conditions and mediated pathways through job appraisals. Findings suggest that well-being is determined not only by one’s exposure to conditions in the work environment but also by the appraisal of these conditions, which helps to reiterate that adopting a methodology that sufficiently captures the complexities of the interactions between police officers’ perceptions and their environment is crucial in determining well-being. The research adds to existing literature on police stress and provides a springboard for developing future research using similar methodologies.

A limitation of the research is the use of self-report measures as the primary data source. While self-report measures are quick, easy to distribute, and are considered reasonable methods of assessing beliefs, feelings, and behaviors, they are also open to biases in reporting. For instance, the police officers who participated in the surveys may have underestimated or overestimated their perceptions in response to the items. Using self-report measures may also inflate the relationship between variables and result in common method variance (CMV). However, CMV was unlikely, as this would have consistently resulted in high correlations, but this was not observed among the variables in the research. Similarly, the absence of a global profile of effects suggests that well-being did not reflect a “complaining culture”. The addition of objective measures is highly desirable but difficult in studies of this type.

In addition, given some of the complexities of the relationships that were examined, more sophisticated analyses using structural equation modelling (SEM) may be useful in future research. However, some experts argue that although there are some advantages to SEM, it is not necessarily better or more appropriate than OLS regressions, particularly when testing mediation and moderation models.

Another limitation is the absence of detailed analyses of demographic variables such as gender. These variables were adjusted for in the models but were not the focus of the analyses. The disparity between the number of male and female officers in this study is not unusual, given the typical distribution within police organizations. While it was not the remit of this study to conduct gender difference analysis, as the focus was on establishing direct and mediated relationships, it may be useful for future studies to consider gender differences in those associations. Another feature that can be rectified in future studies is the cross-sectional design. Longitudinal surveys can be conducted to provide a better indication of causal mechanisms.

## 5. Conclusions

Police work, particularly in developing countries like Jamaica, is highly stressful. Therefore, understanding the underlying mechanisms that may lead to poor health or well-being outcomes among police officers is necessary to help mitigate the consequences and provide early interventions. Using the DRIVE model of work stress as the framework, both direct and indirect effects of work conditions on well-being and health outcomes were observed. From a practical point of view, the findings of this study provide essential evidence that gives credence for periodically monitoring and auditing perceptions of stress and job satisfaction as they are likely precursors to subsequent health problems. Based on these periodic observations, efforts can be made to address antecedent factors that may ultimately affect health over a prolonged period. The specific pathways leading to “negative” psychological outcomes versus those for “positive” outcomes also provide insights into how these effects may occur and should be further explored.

## Figures and Tables

**Figure 1 behavsci-14-00001-f001:**
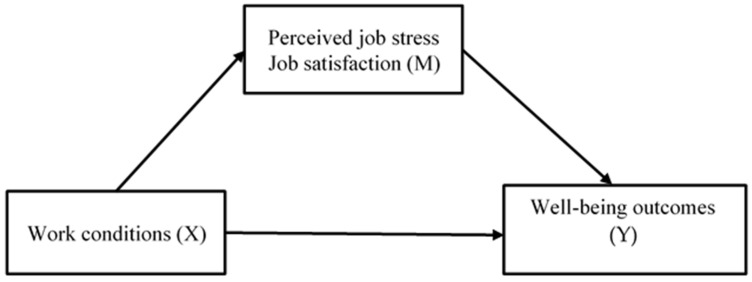
Direct and indirect relationships tested in the mediation analyses.

**Table 1 behavsci-14-00001-t001:** Demographic description of the sample.

Gender (*N* = 578)	Male	*N* = 427	74%
	Female	151	26
Age (*N* = 578)	≤28	200	35
	29–35	197	34
	36+	181	31
Education (*N* = 574)	Secondary	330	58
	Diploma	117	20
	Associate degree	23	4
	Bachelor’s	99	17
	Master’s	5	1
Relationship status (*N* = 578)	Single	122	21
	In a relationship	257	45
	Married	164	28
	separated/divorced/ widowed	35	6
Rank (*N* = 578)	Constable	362	63
	Corporal	128	22
	Sergeant	62	11
	Inspector	26	4
Years of service (*N* = 578)	≤5	246	43
	6–12	157	27
	13+	175	30

**Table 2 behavsci-14-00001-t002:** Hierarchical regression for psychological distress, positive well-being, and general health regressed against demographic and work factors.

Independent Variables	Perceived Job Stress		Job Satisfaction		Psychological Distress		Positive Well-Being		General Health	
	Step 1 β	Step 2 β	Step 1 β	Step 2 β	Step 1 β	Step 2 β	Step 1 β	Step 2 β	Step 1 β	Step 2 β
Demographics										
Gender	0.02	0.03	−0.04	−0.07	0.00	0.01	0.06	0.04	0.00	−0.03
Rank	−0.03	−0.04	0.16 **	0.14 **	−0.07	−0.08	0.10	0.09	−0.01	−0.02
Relationship status	0.04	0.05	−0.07	−0.08 *	0.03	0.05	−0.04	−0.04	0.01	0.00
Years of service	0.10	0.15 **	0.07	−0.02	−0.07	−0.01	0.11	0.06	0.17 **	0.12 *
Work factors										
Neg. work conditions		0.38 ***		−0.29 **		0.36 ***		−0.19 **		−0.24 **
Work support		−0.05		0.25 ***		−0.14 ***		0.20 ***		0.06
Pos. work conditions		−0.09 *		0.20 ***		−0.08		0.11 **		0.12 **
Victimization		0.13 ***		−0.12 **		0.10 *		−0.06		−0.13 **
R^2^	0.01	0.22	0.05	0.33	0.02	0.23	0.04	0.17	0.03	0.15
ΔR^2^		0.21 ***		0.28 ***		0.21 ***		0.13 ***		0.13 ***

Gender: male = 0, female = 1. Rank: constable = 0, above constable = 1. Relationship status: 0 = not in a relationship, 1 = in a relationship. Victimization: ≤1 incident = 0, 2 or more incidents = 1. * *p* < 0.05, ** *p* < 0.01, *** *p* < 0.001.

**Table 3 behavsci-14-00001-t003:** Model summary of indirect effect of work factors on psychological distress through job stress and job satisfaction.

			Perceived Job Stress	Job Satisfaction
	Total Effects	Direct Effects	Indirect Effects	Indirect Effects
Negative Work conditions	0.24 ***	0.21 ***	0.02, CI [0.00, 0.05] ^a^	0.01, CI [−0.01, 0.03]
Work support	−0.10 ***	−0.09 **	−0.00, CI [−0.01, 0.01]	−0.01, CI [−0.03, 0.01]
Positive Work conditions	−0.10	−0.08	−0.01, CI [−0.03, −0.00] ^a^	−0.01, CI [−0.04, 0.01]
Victimization	1.13 *	0.95 *	0.13, CI [0.01, 0.34] ^a^	0.05, CI [−0.07, 0.25]

Note: CI, confidence intervals. * *p* < 0.05. ** *p* < 0.01. *** *p* < 0.001. ^a^ significant indirect effect.

**Table 4 behavsci-14-00001-t004:** Model summary of indirect effect of work factors on positive well-being through job stress and job satisfaction.

			Perceived Job Stress	Job Satisfaction
	Total Effects	Direct Effects	Indirect Effects	Indirect Effects
Negative job				
characteristics	−0.21 ***	−0.16 ***	−0.01, CI [−0.04, 0.03]	−0.05, CI [−0.09, −0.01] ^a^
Work support	0.24 ***	0.20 ***	0.00, CI [−0.00, 0.01]	0.04, CI [0.01, 0.08] ^a^
Positive job				
characteristics	0.24 **	0.18 *	0.00, CI [−0.02, 0.02]	0.06, CI [0.02, 0.13] ^a^
Victimization	−1.09	−0.73	−0.03, CI [−0.25, 0.20]	−0.33, CI [−0.77, −0.08] ^a^

Note: CI, confidence intervals. * *p* < 0.05. ** *p* < 0.01. *** *p* < 0.001. ^a^ significant indirect effect.

**Table 5 behavsci-14-00001-t005:** Model summary of indirect effect of work factors on general health through job stress and job satisfaction.

			Perceived Job Stress	Job Satisfaction
	Total Effects	Direct Effects	Indirect Effects	Indirect Effects
Negative work				
conditions	−0.06 ***	−0.04 ***	−0.01, CI [−0.02, −0.00] ^a^	−0.01, CI [−0.02, −0.00] ^a^
Work support	0.02	0.01	0.00, CI [−0.00, 0.00]	0.01, CI [0.00, 0.02] ^a^
Positive work conditions	0.06 **	0.04 *	0.00, CI [0.00, 0.01] ^a^	0.01, CI [0.01, 0.03] ^a^
Victimization	−0.56 **	−0.44 *	−0.05, CI [−0.13, −0.01] ^a^	−0.08, CI [−0.17, −0.02] ^a^

Note: CI, confidence intervals. * *p* < 0.05. ** *p* < 0.01. *** *p* < 0.001. ^a^ significant indirect effects.

## Data Availability

Contact K.N.—knelson@utech.edu.jm. The research described in this paper was part of the PhD thesis submitted by Dr Nelson. A digital copy of this unpublished thesis is in the Cardiff University repository: K. Nelson. Behind the frontlines: occupational stress and well-being in Jamaican police officers. Commonwealth Studentship. Awarded 2017. http://orca-mwe.cf.ac.uk/99877/.

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
