# Peer review of "Psychosocial Work Conditions as Determinants of Well-Being in Jamaican Police Officers: The Mediating Role of Perceived Job Stress and Job Satisfaction"

_behavsci, 2023, doi:10.3390/bs14010001_

Round 1

Reviewer 1 Report (Previous Reviewer 3)

Comments and Suggestions for Authors

Thank you for your effort revising the manuscript and addressing the issues I raised. I do like the current edition. 

Author Response

We thank the reviewer for the comments during the review process.

Reviewer 2 Report (New Reviewer)

Comments and Suggestions for Authors

Thank you a very clearly written paper, apposite and situated within contemporary research. The rationale for the work is clear and it has important implications for those working in the Jamaican police force, it is likely generalisable more universally in policing. The only issues I identified were typographical please see below:

Line 165 (p.4) there is a reference number missing from the parentheses.

Line 329 (p.7) It says 31% were 36. It should say 36 (or over)

In table 2 next to positive work conditions there is a decimal point missing, it presently says 20*** think it should be .20***

In table 3 can authors check victimisation, it says 1.13* in table

In table 4 can authors check victimisation again it says -1.09.

Author Response

The typographical errors have been corrected. The total effects in tables 3 and 4 are greater than 1 as they reflect the sum of the direct and indirect effects.

Reviewer 3 Report (New Reviewer)

Comments and Suggestions for Authors

This is a well executed cross-sectional questionnaire study of the conditions for police in Jamaica. I tried to find information regarding non-participation but did not find any. This should be added if I am right in my non-detecting this information.

I also find that the authors do not discuss sufficiently the problems arising due to the fact that it is a cross-sectional study using self-rated information in self-administered questionnaires. Yes the authors do argue that there is little evidence of inflated correlations but the readers should still get more on this.

I was quite interested in the lack of indirect effects in some of the most crucial associations. That does strengthen the argument that effects of wellbeing are not confined to "complainers" and this main result should be used in discussions with employers!

Author Response

We thank the reviewer for the useful comments.

Information on recruitment has now been provided which shows the problem of assessing response rate.

Further information is also provided on the problems of cross-sectional studies with subjective reports.

The absence of support for an explanation based on "complainers" is also discussed.

This manuscript is a resubmission of an earlier submission. The following is a list of the peer review reports and author responses from that submission.

Round 1

Reviewer 1 Report

Comments and Suggestions for Authors

 I made the next suggestions and questions to the authors. I hope these will be useful to improve the article.

Methodology

Participants

Just describe the inclusion and no inclusion criteria for the participants in the study.

Information about the participants (demographic and the Table 1) should be moved to the first part in the results.

Create a data analysis where the information written in lines 273-278 were placed at the end of the methodology.

Results

Data analysis is not results, staring this section with the information written in line 279.

Factor analysis. In lines 290-293 authors mention PCA revelated three components. I would like to see the eigenvalues and the variance explained. Add a table with this information.

Did the authors a reliability analysis using at least Cronbach Alpha?

For this kind of studies (factor analysis and structural), it is necessary to include a diagram showing the relationship between the constructs, direct, indirect, and total effects.

Finally, the most important issue is the references. Most of the references are “old” and the average age of the references is 17 years (max 47 and min 6 years). I understand the citation of Prof. Karasek (1979), but the text must be supported with new studies. My recommendation is to improve the manuscript using more recent references.

Author Response

See file

Reviewer 2 Report

Comments and Suggestions for Authors

In the attached file.

Author Response

See file

Reviewer 3 Report

Comments and Suggestions for Authors

I really enjoyed reading the paper. Authors presented a very interesting phenomenon with care and compassion. To help improve the coherence of the story, I'd like to share some of my observations as listed below in more details: 

Structure the writing with a convention logic: You did a good job laying out the research questions and relevance of the study (first part of the introduction). The next steps should be: 

- First, clearly define the key variables: you often use different terminology for the same variable and what you measured is not what you talk about in the hypotheses development section. Specifically: 

 The IV, work conditions. You use multiple terms throughout the paper. For instance, psychological work conditions, nature of organization, organizational administration and structure, organizational factors, perceived work conditions, work conditions, conditions in the police work environment, DCS etc. From line 89 - 97, you seem to differentiating two types of organizational factors and specifically indicate that organizational administration and structure are more important than operational experiences in predicting well-being. Yet, for the rest of the paper, you mostly focus on the experiences part and actually use an experience scale  (victimization) to operationalize the IV, as the psychological work conditions. 

If your story is about psychological factors, then be specific and consistent from beginning to the end. 

- Second, what is the overarching theoretical framework (a specific theory or theories) are you going to use to explain/predict the relationships between variables?

Citing previous empirical findings and evidence is not the same as developing theoretical predictions. And yet, you did use the term "theoretical model" (Lind 113) - but you did not explain what these theories would say about the relationships of the variables. Rather you cited more findings of different studies. As a reader, I cannot see the apparent reasoning behind your hypothesis. 

- Finally, if this is an empirical paper (which I assume this paper is), the logic is that, theory or theories predict a relationship between IV, DV, and/or Mediators, that leads to how you hypothesize the relationships (i.e., IV-DV; IV-Mediator1 and 2 - DV). Then you will collect data to test your predictions to see if they are supported. But what you have for all the three hypothesis are laid out like this: so and so's studies have found evidence of the relationship(s), our study support the relationships, that leads us to hypothesize the relationships. The word "support" comes before your hypotheses makes me think you have the data first, then build a story around your data. 

Comments on the Quality of English Language

Overall quality is good. But more editing needed in some sections.

Author Response

see file

Round 2

Reviewer 3 Report

Comments and Suggestions for Authors

Thank you for your responses. Based on the track-marked manuscript, the only change in the writing (not the citation changes) is to add two sentences on page 2 (Line 88 - 93). Unfortunately, this addition did not address my comments sufficiently. My original comments and suggestions stand.

Comments on the Quality of English Language

Much improved but need more edit. 
